# Quartz Crystal Microbalance Measurement of Histidine-Rich Glycoprotein and Stanniocalcin-2 Binding to Each Other and to Inflammatory Cells

**DOI:** 10.3390/cells11172684

**Published:** 2022-08-29

**Authors:** Tor Persson Skare, Hiroshi Kaito, Claudia Durall, Teodor Aastrup, Lena Claesson-Welsh

**Affiliations:** 1Science for Life and Beijer Laboratories, Department of Immunology, Genetics and Pathology, Uppsala University, Dag Hammarskjöldsv 20, 751 85 Uppsala, Sweden; 2Attana AB, Greta Arwidssons Väg 21, 114 19 Stockholm, Sweden

**Keywords:** histidine-rich glycoprotein, stanniocalcin-2, protein complex, inflammatory cells, quartz crystal microbalance

## Abstract

The plasma protein histidine-rich glycoprotein (HRG) is implicated in the polarization of macrophages to an M1 antitumoral phenotype. The broadly expressed secreted protein stanniocalcin 2 (STC2), also implicated in tumor inflammation, is an HRG interaction partner. With the aim to biochemically characterize the HRG and STC2 complex, binding of recombinant HRG and STC2 preparations to each other and to cells was explored using the quartz crystal microbalance (QCM) methodology. The functionality of recombinant proteins was tested in a phagocytosis assay, where HRG increased phagocytosis by monocytic U937 cells while STC2 suppressed HRG-induced phagocytosis. The binding of HRG to STC2, measured using QCM, showed an affinity between the proteins in the nanomolar range, and both HRG and STC2 bound individually and in combination to vitamin D3-treated, differentiated U937 monocytes. HRG, but not STC2, also bound to formaldehyde-fixed U937 cells irrespective of their differentiation stage in part through the interaction with heparan sulfate. These data show that HRG and STC2 bind to each other as well as to U937 monocytes with high affinity, supporting the relevance of these interactions in monocyte/macrophage polarity.

## 1. Introduction

Histidine-rich glycoprotein (HRG) is a 75 kDa abundant plasma protein produced by hepatocytes and implicated in cancer immune responsiveness [1]. HRG is organized as a multi-domain structure of two N-terminal cystatin-like domains, followed by a histidine–proline-rich (His/Pro-rich) domain containing 12 pentapeptide repeats of Gly-His-His-Pro-His. The His/Pro repeats, which are highly conserved among mammalian species [2], are flanked by two Pro-rich regions and a C-terminal domain [3]. The cystatin domains have been implicated in HRG’s antibacterial effects [4] and in IgG and complement C1q binding [5]. The His/Pro-rich domain binds heparan sulfate in a Zn^2+^ dependent manner [6]. This domain is also critical for the anti-angiogenic properties of HRG [7]. The structure of HRG’s NH_2_-terminal domain has been solved, indicating a redox-regulated release of the NH2-terminal domain from the His/Pro-rich domain [8]. HRG has also been classified as an intrinsically unstructured protein, unable to attain an ordered or fixed conformation [9].

HRG’s multi-domain organization allows for interactions with a range of proteins, both intracellular such as tropomyosin, extracellular such as stanniocalcin 2 (STC2), and proteins participating in the coagulation cascade, including plasminogen, plasmin and fibrinogen [10,11]. Consequently, HRG is involved in diverse processes including defense against bacterial infections, regulation of coagulation and fibrinolysis, inflammation and angiogenesis. Thus, HRG exerts antibacterial effects and may serve as a clinical biomarker for sepsis [12]. Moreover, HRG accelerates both coagulation and fibrinolysis in a *Hrg^–/–^* mouse model [13]. Rare familiar cases of HRG deficiency support the role of HRG in the regulation of coagulation [14]. Certain HRG effects are dependent on changes in gene regulation in monocytes/macrophages, promoting a phenotypic switch towards anti-tumor immunity and dampened tumor growth and metastasis [1]. While HRG administration to tumor-bearing wildtype mice results in suppressed tumor growth and metastasis, tumor growth is accelerated in *Hrg^–/–^* mice, and tumor macrophages are predominantly of an M2 phenotype in these *Hrg*-deficient mice [1,7,15].

We previously investigated the immunomodulatory role of HRG on inflammatory cells and found that HRG appears in complex with STC2 and that HRG modulates STC2-mediated gene regulation on U937 monocytic cells [16]. STC2 is a glycosylated homodimeric protein expressed in the placenta, endothelial cells, fibroblasts and cardiomyocytes [17]. In mouse glioma, tumor-infiltrating leukocytes express STC2 [16]. Like HRG, STC2 is involved in inflammatory processes and in Ca^2+^ and PO_4_ homeostasis [18,19]. *Stc2^–/–^* mice show decreased overall growth, suggesting an important role for STC2 in muscle and bone development [20].

To explore the biochemical properties of HRG, STC2 and the complex of the two, purified proteins were analyzed using a novel quartz crystal microbalance (QCM) technique. Through the QCM analyses, the affinity and kinetics of the HRG-STC2 interaction as well as the binding of HRG and STC2 to the U937 monocyte cell surface were tested [21]. U937 is a human histiocytic lymphoma cell line, which differentiates towards a macrophage phenotype after treatment with vitamin D3 (vitD3) [22]. The functionality of the HRG and STC2 recombinant protein preparations were assessed by the induction of phagocytosis by differentiated U937 monocytes. While HRG administration stimulated phagocytosis, STC2 abrogated this HRG-dependent effect. In the QCM system, vitD3 differentiated live U937 cells, but not undifferentiated cells, bound both HRG and STC2, with an affinity in the micromolar range. HRG, but not STC2, also bound to fixed U937 cells, possibly representing HRG interactions with heparan sulfate exposed through the fixation. Thus, digestion with heparinase revealed high affinity binding sites for HRG on the differentiated, fixed U937 cells. Combined, these findings confirm the specificity of the HRG-STC2 interaction and show high affinity binding sites on the monocytic cell surface.

## 2. Materials and Methods

### 2.1. Differentiation of U937 Cells

The human histiocytic lymphoma cell line U937 [23] (American Type Culture Collection, ATCC 1593, RRID:CVCL_0007) was a kind gift from Professor Kenneth Nilsson, Uppsala University. The cells were cultured in an RPMI 1640 medium supplemented with 10% fetal bovine serum (FBS) and 1% penicillin/streptomycin (cat. no. 61870036; Life Technologies, Grand Island, NY, USA). For monocyte differentiation, U937 cells were incubated in 10 nM 1α,25-Dihydroxyvitamin D3 (vitD3; cat. no. 17936, Merck Life Science, Darmstadt, Germany) for 15 h, centrifuged (1500 rpm, 5 min), and resuspended in fresh medium. Differentiation was determined by real-time reverse transcriptase-PCR (qPCR) to detect CD14 transcripts. The mRNA was extracted from cells using the RNAeasy mini kit (Qiagen, Germantown, MD, USA) and RNA was reverse transcribed with iScript adv (cat. no. 1725038, Bio-Rad, Hercules, CA, USA). Gene expression was determined using TaqMan universal master mix (cat. no. 4304437, Life Technologies, Grand Island, NY, USA) in the CFX96 Real-Time PCR Detection System (Bio-Rad, Hercules, CA, USA) with TaqMan primers against human CD14 (cat. no. Hs 00169122, Life Technologies, Grand Island, NY, USA) and human GAPDH (cat no. 4352934, Applied Biosystems, Waltham, MA, USA). Cycle threshold values were calculated with CFX Maestro v. 1.1 software (Bio-Rad, Hercules, CA, USA).

### 2.2. Purified Proteins and Phagocytosis Assay

The U937 cells were seeded at 10^4^ cells per well in 8-well chamber slides (cat. no. 80826, ibidi, Fitchburg, WI, USA). At the start of the experiment, cells were incubated with 10 nM vitD3, recombinant, in-house purified HRG (mouse) at 1 µg/mL (13.3 nM) [24], and STC2 (mouse) (cat. no. STC2-16118 M, Creative Biomart, Shirley, NY, USA) or inactive HRG protein at equivalent molar concentrations together with sterile green E. coli bioparticles (cat. no. 4616, Essen Bioscience, Ann Arbor, MI, USA) at 33 µg/mL. Following 20 h incubation at 37 °C in 5% CO_2_, cells were imaged at 10× with a Zeiss LSM 700 Microscope with AxioCam HRm and Zen Black software (Zeiss, Oberkochen, Germany). Quantification was completed by counting of fluorescent cells in relation to all cells per image, using ImageJ (NIH).

### 2.3. Co-Immunoprecipitation and Immunoblotting

Equimolar concentrations of active or inactive HRG and STC2 were incubated on ice for 30 min followed by incubation with an anti-STC2 antibody (cat. no. hpa045372; Merck Life Science, Darmstadt, Germany) for 1 h. Protein G Sepharose (cat. no. 71708300 AM, GE Healthcare, Chicago, IL, USA) was added and incubated at 4 °C for 1 h. Following centrifugation and washes, samples were heated at 97 °C for 3 min for dissociation. Samples were separated by SDS-PAGE, transferred to a PVDF membrane (Millipore, Burlington, MA, USA), blocked in blocking buffer (5% milk in Tris-buffered saline and 0.1% Tween20) for 1 h and incubated with primary anti-human HRG antibody raised in rabbit (#0119) [7] overnight at 4 °C. Membranes were washed and incubated with HRP-conjugated secondary anti-rabbit antibody (Life Technologies, Grand Island, NY, USA) in blocking buffer for 1 h at room temperature. The development was performed with ECL prime (GE Healthcare, Chicago, IL, USA) and the luminescence signals detected using ChemiDoc MP (Bio-Rad; Hercules, CA, USA). Next, membranes were re-incubated with the STC2 antibody (cat. no. hpa045372, Merck Life Science, Darmstadt, Germany) overnight at 4 °C and developed again as described above.

### 2.4. Binding of HRG to STC2

Low noise block (LNB) chips were pre-wet with HEPES-buffered Steinberg’s solution (HBS-T) and inserted into an Attana Cell^TM^200 instrument. When the signal was stabilized (<0.2 Hz/min), STC2 or HRG protein (50 µg/mL) were immobilized on the surface with a flow rate of 10 µL/min at 22 °C using the amine coupling kit. Different concentrations of HRG (3.12, 6.25 and 12.5 µg/mL (each in triplicate)) and STC2 (7.5, 15 and 30 µg/mL) were injected after blank injections (phosphate-buffered saline; PBS) followed by regeneration injections at pH 1 for 30 s. PBS was used as the running buffer for the blank injections and to dilute HRG. Glycine (10 mM, pH 1) was used as the regeneration buffer. The biochemical assay was carried out at a flow rate of 10 µL/min, at 22 °C and with a 500 s dissociation time. The data were prepared by subtracting the blank injections from the HRG injections using the Attana evaluation software (version 3.5.0.7, Attana AB Stockholm, Stockholm, Sweden). The curve fitting was performed with Tracedrawer (Ridgeview Instruments AB, Uppsala, Sweden), using the 1:1 global interaction model. The number of independent experiments (mostly 3) performed are given in the figure legends.

### 2.5. Binding of HRG and STC2 to U937 Cells Treated or Not with Heparinase and Fixative

LNB-CC chips were pre-wet with HBS-T and then inserted into an Attana CellTM 200 (Attana AB, Stockholm, Sweden). When the signal was stabilized (<0.2 Hz), lectin (50 µg/mL) was coupled by amine coupling. Cells were then seeded at a density of 2 × 105 cells per chip and left to settle for 45 min at room temperature. After seeding, the cells were washed with PBS, stained with Hoechst 33342 solution for 15 min, and washed three times, followed by imaging using a fluorescence microscope (Nikon Eclipse 80i, Minato City, Tokyo, Japan). Next, the chips were inserted in the instrument (Attana CellTM 200, Stockholm, Sweden) and left to equilibrate (<0.2 Hz/min) under flow (RPMI 1640 medium, 20 μL/min at 37 °C). STC2, HRG or a mix of the two (10 μg/mL) were injected manually over the cells and the responses were recorded for 30 min. Between injections, chip surfaces were regenerated using Glycine (10 mM pH 1) for 50 s for experiments involving fixed cells.

When indicated, cells were treated with heparinase using a mixture of heparinase-I, -II, and -III (IBEX Pharmaceuticals, Montreal, QC, Canada), which was added to the cultures to a final concentration of 3.4 mU/mL for each enzyme for 1 h at 37 °C before seeding the cells on the activated chips. The fixation of cells was performed just after seeding by removing the PBS and adding 50 µL of a 3.7% formaldehyde solution. Subsequently, the chips were incubated at 4 °C for 10 min followed by washing with PBS three times. The data were prepared by subtracting the blank injections from the analyte injections using the Attana evaluation software (version 3.5.0.7, Attana AB Stockholm, Stockholm, Sweden). The signal output is given in frequency (Hz) and is directly related to changes in mass on the sensor surface. The negative changes of resonance frequency are depicted. The curve fitting was performed with Tracedrawer (Ridgeview Instruments, Uppsala, Sweden), using the 1:1 or 1:2 binding models (only one component reported) and global interaction model. At least two independent experiments were performed with 3 technical repeats within each experiment.

## 3. Results

### 3.1. HRG Increases Phagocytosis by U937 Monocytes

First, the bioactivity of purified, recombinant STC2 and HRG preparations were determined. To ensure that the recombinant proteins could form a complex, as previously shown by co-immunoprecipitation from co-expressing cells [16], antibodies against STC2 were used for pull-down from a mixture of the two proteins, followed by immunoblotting (Figure 1A; see Appendix A for uncropped blots). In parallel, a preparation of HRG serendipitously denatured during purification, and was used as a negative control (“inactive HRG”). Active HRG was efficiently co-immunoprecipitated with STC2 while the inactive HRG was only inefficiently pulled down by STC2 (Figure 1B). Still, the inactive HRG was detected by the polyclonal anti-HRG antibody upon immunoblotting, ensuring that this preparation indeed consisted of HRG. The inactive HRG-preparation was used as a negative control in subsequent experiments.

### 3.2. HRG and STC2 Interact with Nanomolar Affinity

HRG and STC2 are both secreted proteins, and therefore, it is not immediately obvious how they exert their modulatory effects on inflammatory cells. To investigate the interactions of the individual proteins and the complex with each other and with cells, we employed QCM technology [25], which allowed for real-time and label-free evaluation of the protein interactions in both a cell-free and cellular environment.

First, STC2 was immobilized on the QCM chip surface (Figure 2A). Next, the binding of increasing concentrations of HRG to the STC2-coated chip was analyzed using a kinetic 1:1 global interaction model, which showed an association rate constant (K_a1_) of 2.6 × 10^4^ M^−1^·s^−1^ and dissociation rate constant (K_d1_) of 1.4 × 10^−3^ s^−1^, resulting in an estimated binding affinity of 55 nM between HRG and STC2 (Figure 2B). In contrast, the inactive HRG failed to bind to the immobilized STC2 (Figure 2C). Moreover, when HRG was immobilized on the grid, STC2 was not retained (Figure 2D). This result indicates that the interaction between HRG and STC2 may be dependent on HRG’s conformation or that a binding pocket in HRG, involved in the retention of STC2, was compromised when HRG was immobilized onto the chip. HRG is classified as an intrinsically unstructured protein [9], i.e., a protein that lacks a fixed three-dimensional structure, and it may therefore be structurally less stable and, in particular, not retain a more complex binding epitope upon immobilization.

### 3.3. Live U937 Cells Bind HRG after vitD3 Differentiation

Next, we assessed binding of HRG, STC2 and the complex to live U937 cells, vitD3-differentiated or not (Figure 3A). First, we confirmed the ability of U937 cells to differentiate to monocytes on the chip surface in response to vitD3. Relative CD14 expression increased >1000-fold after vitD3 treatment, ensuring that the cells indeed had differentiated to monocytes in response to vitD3 (Figure 3B). This is in agreement with previously reported effects of HRG treatment on CD14 expression in vitD3-induced U937 cells [15]. Binding of both HRG and STC2 to the undifferentiated U937 cells was low and binding increased markedly when cells were induced to differentiate by vitD3 treatment (Figure 3C–F). The association rate constants of HRG and STC2 binding tested individually (Figure 3C–F) or in combination (Figure 3G,H) increased slightly with differentiation, in keeping with the increased expression of a specific binding protein(s) expressed on the U937 surface in response to the vitD3 treatment. When a mixture of HRG and STC2 was tested on the differentiated U937 cells, the binding affinity remained in the µM range, similar to that recorded for the individual proteins (Figure 3G,H).

We conclude that both HRG and STC2 bound to U937 monocytes with affinities in the µM range and with an estimated 1:1 interaction mode both for the individual proteins and the mixture.

### 3.4. Binding of HRG to Fixed U937 Cells Is Independent of vitD3-Induced Differentiation

Due to the potential challenge in separating interaction properties from interaction-induced changes in the live U937 monocytes immobilized on the QCM chip surface, cells were then fixed, and the binding of HRG and STC2 was determined. As expected, inactive HRG, used as a negative control, displayed no interaction with U937 cells, differentiated or not, even at high concentrations (100 µg/mL) (Figure 4A). In addition, STC2 failed to bind both to undifferentiated and differentiated cells in a specific manner, with the response dropping down to baseline immediately after the end of injection (Figure 4B). In contrast, bioactive HRG interacted with the fixed cells with an affinity around 3.5 nM to undifferentiated cells and 166 nM to differentiated cells (Figure 4C,D) with a 1:1 interaction mode. In combination, these data show very different properties for STC2 and HRG interactions with fixed cells, as fixation exposed binding sites for HRG on undifferentiated cells were recorded while not observed with live cells. However, we cannot categorically exclude that the binding sites exposed upon fixation, at least in part, could represent intracellular ligands for HRG [3], either proteinase K-resistant or -sensitive.

Instead, we hypothesized that these binding sites for HRG on undifferentiated, fixed cells may involve heparan sulfate epitopes [6]. To investigate this possibility further, cells were treated with heparinase before HRG binding. The treatment with heparinase changed the dissociation curve to a 1:2 interaction mode [26]. The affinities for the two categories of HRG binding to heparinase-treated undifferentiated cells were 88 nM and 0.84 µM, respectively (Figure 4E). Heparinase-digestion of differentiated U937 cells on the other hand revelated affinities for HRG of 0.03 nM and 0.4 µM, respectively (Figure 4F). We suggest that this non-heparan sulfate-dependent, high-affinity binding of HRG to fixed, differentiated cells may represent binding to a molecular entity responsible for transducing the biological effects of HRG, such as increased phagocytosis.

## 4. Discussion

This study aimed to understand the binding properties of the soluble plasma proteins HRG and STC2, motivated by their induction of gene regulatory programs that steer the phenotype of the monocyte/macrophage towards pro- or anti-inflammatory activities. We have previously shown that HRG, either administered as a recombinant protein, overexpressed by tumor cells, or delivered through adenovirus-mediated gene therapy, polarizes monocytes/macrophages to an anti-tumor immune profile, allowing for the recruitment of cytotoxic T cells to the tumor [1,16,24]. The effect of HRG is accompanied by tumor vessel normalization and suppressed tumor growth [1,27]. In a screen to identify HRG binding partners on the surface of monocytes mediating HRG’s gene regulatory effects, STC2 was identified and shown to have broad effects on inflammatory gene regulation [16]. Here, we asked whether HRG and STC2 bind to a common cell surface molecule (“receptor”) to steer monocyte gene regulation in a concerted action, based on the characteristics of binding to U937 monocytic cells.

To explore the binding properties of HRG and STC2, we employed a QCM biosensor methodology using unmodified, label-free proteins. This is important as modifications such as fluorescent peptide linkers or fusion partners, e.g., green fluorescent protein, can affect the folding of the modified protein and cause unnatural protein interactions. Moreover, radioactive labelling of proteins to determine protein interactions can harm the protein through the harsh methods used to introduce the label. In QCM, a thin quartz crystal disk is sandwiched between two electrodes. Changes in mass, e.g., upon binding of HRG to the surface of immobilized cells, results in a mechanical deformation of the disk, which mediates a frequency change in the quartz crystal that is proportional to the change in mass, which allows for calculations of affinity [28,29]. The technology also allows for real-time tracking of the kinetics of the interaction. However, the application of live cells, in particular the issues with non-adherent cells undergoing a differentiation process, are not trivial for QCM analyses. The cells need to remain on the grid, unaffected by the flow of the medium and pursue the differentiation program. By measuring CD14 expression, we could show that the U937 cells on the grid indeed could be induced by vitD3 to differentiate towards the monocyte lineage. However, as live cells may react to the binding component, in this case, HRG and STC2, kinetic measurements may represent both interaction properties and interaction-induced cell surface modifications. We, therefore, analyzed the binding of HRG and STC2 also to fixed cells. The utilization of fixed cells allowed for repeated measurements and different concentrations of analyte on the same cell surfaces. This increases reproducibility and leads to more robust data. However, fixation may interfere with protein–protein interactions and affect binding. The formaldehyde fixation utilized here is a preferred fixative for preserved immunoreactivity [30]. Unexpectedly, upon fixation of the cells, STC2 failed to bind to cells irrespective of the differentiation stage, while HRG bound to fixed cells both with and without vitD3 treatment (Figure 4). We therefore suggest that STC2 and HRG bind to distinct molecular entities on the U937 cells, and that the STC2 interactive surface was denatured upon fixation, however, definite proof for this assumption requires identification of the binding surfaces for HRG and STC2.

The binding of HRG to undifferentiated, fixed U937 cells may be due to fixation-induced exposure of heparan sulfate. HRG is known to bind with heparan sulfate in a Zn^2+^-dependent manner, and this interaction is required for the anti-angiogenic effects of HRG [6]. We addressed the role of heparan sulfate by incubating cells with heparinase. Although incomplete, as revealed by the remaining binding of HRG to undifferentiated U937 cells, the digestion resulted in a change of the interaction mode from a 1:1 model with linear dissociation curves to a 1:2 model. This change supports the hypothesis that HRG binds to two classes of binding epitopes with different affinities—a lower affinity heparan sulfate-dependent binding and a higher affinity binding epitope that we hypothesize may represent the signal transducing cell surface expressed receptor mediating the gene regulatory effects of HRG. Further studies include determining whether the interaction between STC2 and HRG is Zn^2+^-dependent, to identify, in each protein, the minimal binding stretch for their interaction and ultimately, to identify the cell surface expressed binding proteins/receptors for HRG and STC2.

## 5. Conclusions

The main findings from this study are (1) HRG and STC2 promote distinct U937 differentiation programs as STC2 suppressed the HRG-induced increase in phagocytosis. Moreover, (2) HRG and STC2 bind to each other with nanomolar affinity, and the interaction is stable, as evidenced by the relatively slow dissociation rate. Both proteins also (3) bind to differentiated, live U937 cells with µM affinities, i.e., to cells that, in response to vitD3, have initiated a gene regulatory program similar to that accompanying monocyte differentiation marked by the expression of CD14 [31]. It is possible that the binding of both HRG and STC2 to live cells is, in part, heparan sulfate-dependent and that differentiation is required to induce expression of a particular category of heparan sulfated proteoglycans. Finally, (4) while STC2 fails to bind to fixed U937 cells, the high-affinity binding of HRG to heparinase-treated fixed cells may represent the interaction with a specific cell surface receptor mediating the biological effects of HRG. These results indicate that STC2 and HRG interact with distinct rather than shared cell surface binding epitopes.

## Figures and Tables

**Figure 1 cells-11-02684-f001:**
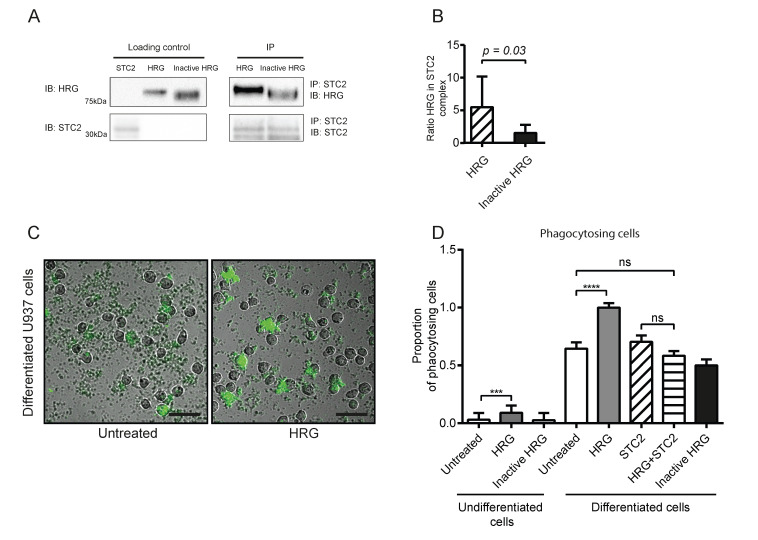
Recombinant HRG binds STC2 and modulates phagocytosis of bioparticles. (**A**) Co-immunoprecipitation of HRG but not inactive HRG with STC2. Recombinant proteins (2 µg each) were separated on SDS-PAGE as individual preparations (loading control) or after mixing and immunoprecipitation (IP) using antibodies against STC2, followed by immunoblotting (IB) as indicated. (**B**) Ratio of HRG (active or inactive) band intensities in the STC2 immunoprecipitate normalized to corresponding active and inactive HRG loading controls. Statistical analysis; Student’s *t*-test. (**C**) Representative microscope images of U937 monocytes without (left) or with treatment with active HRG (right) in the phagocytosis assay. Green cells have engulfed pH-sensitive fluorescent bioparticles. Scale bar; 50 µm. (**D**) Quantification of phagocytosis efficiency in the different treatment conditions. The proportion of positive (green) phagocytotic U937 cells to all cells per field of vision is shown in relation to the positive cells/total cells in the vitD3 differentiated HRG-treated condition (set to 1). Statistical analysis; Tukey’s multiple comparisons test (**D**). *** *p* < 0.001; **** *p* < 0.0001.

**Figure 2 cells-11-02684-f002:**
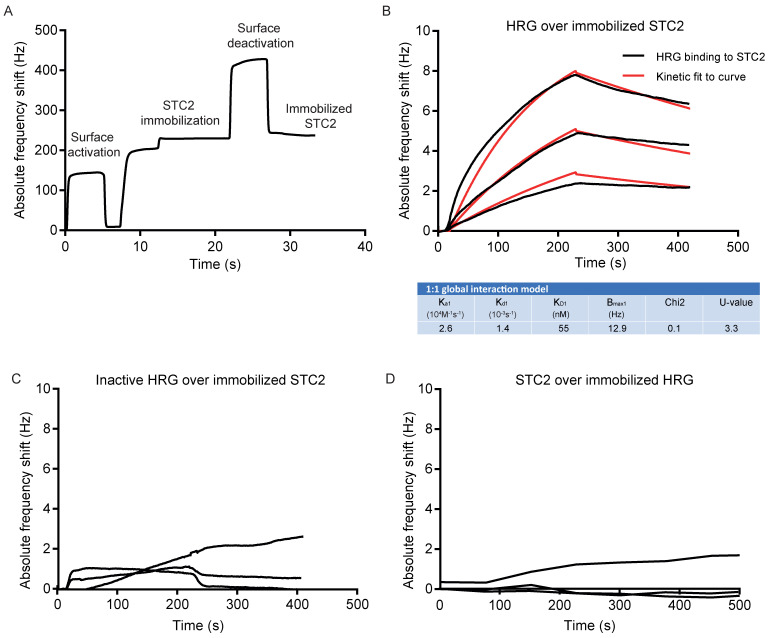
Affinity determination of HRG’s binding to STC2 using QCM. (**A**) The immobilization of STC2 on the QCM LNB sensor surface. (**B**) Sensorgram and kinetic analysis showing chip-immobilized STC2 and binding of HRG at three different concentrations: 50, 100 and 200 nM. Black lines: experimental curves. Red lines: fitted curves. For representative sensorgram shown, three injections per concentration, two independent experiments. (**C**) Sensorgram showing chip-immobilized STC2 and lack of binding of inactive HRG tested at three different concentrations: 50, 100 and 200 nM. For representative sensorgram shown, three injections per concentration, two independent experiments. (**D**) Sensorgram showing chip-immobilized HRG and lack of binding of STC2, tested at three different concentrations: 200 nM, 450 nM and 900 nM. For representative sensorgram shown, two injections per concentration, three independent experiments.

**Figure 3 cells-11-02684-f003:**
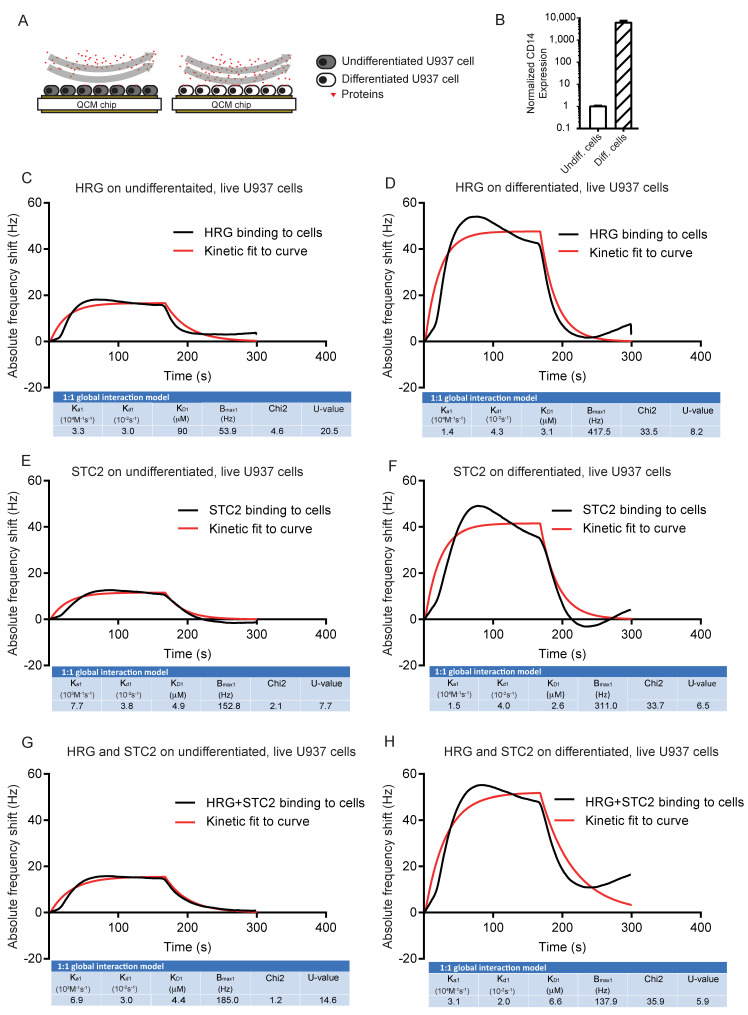
Binding of HRG and STC2 individually and together to live U937 cells. (**A**) Schematic outline of the experimental setup. Undifferentiated or vitD3 differentiated U937 cells, immobilized on QCM LNB chips with HRG, STC2 or a mix of the two, injected over chip surfaces. (**B**) Real-time qPCR data of CD14 expression normalized to GAPDH on undifferentiated and vitD3 differentiated U937 cells seeded on the QCM chip. Three independent analyses. (**C**–**F**) Representative sensorgram showing frequency response from injections over undifferentiated (**C**,**E**,**G**) and vitD3 differentiated (**D**,**F**,**H**) live U937 cells. Black lines: experimental curves. Red lines: fitted curves of the 1:1 interaction model. For representative sensorgrams shown, three injections per concentration, two independent experiments.

**Figure 4 cells-11-02684-f004:**
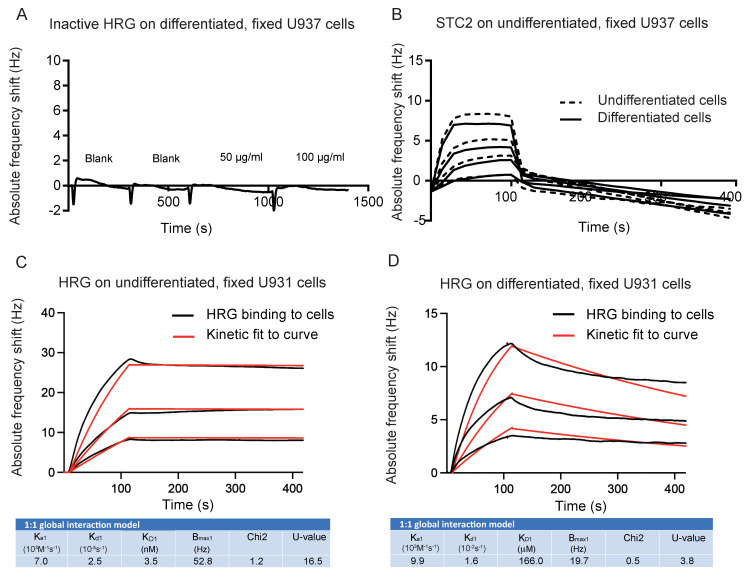
Affinity of HRG for binding to fixed U937 cells. (**A**) Sensorgram showing frequency response to inactive HRG over vitD3 differentiated, fixed U937 cells. (**B**) Sensorgram showing frequency response to four concentrations (125 nM, 250 nM, 500 nM, 1 µM) of STC2 over fixed, undifferentiated (dashed lines) and vitD3 differentiated (straight lines) U937 cells. The mean of two injections is shown. (**C**,**D**) Sensorgram and kinetic analysis show the binding of HRG at three concentrations (25, 50 and 100 nM) to undifferentiated (**C**) or vitD3 differentiated (**D**), fixed U937 cells. Black lines: experimental curves. Red lines: fitted curves of the 1:1 interaction model. For representative sensorgrams shown, three injections per concentration, three independent experiments. (**E**,**F**) Sensorgram and kinetic analysis showing frequency response to three concentrations of HRG (25, 50 and 100 nM) to fixed, undifferentiated (**E**) or vitD3 differentiated (**F**) U937 cells after treatment with heparinase. Black lines: experimental curves. Red lines: fitted curves of 1:2 interaction model. For representative sensorgrams shown, three injections per concentration, three independent experiments.

## Data Availability

Data supporting the results shown in this paper can be obtained from the lead authors (T.A. and L.C.W.) upon reasonable request.

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
