# Peer review of "Quartz Crystal Microbalance Measurement of Histidine-Rich Glycoprotein and Stanniocalcin-2 Binding to Each Other and to Inflammatory Cells"

_cells, 2022, doi:10.3390/cells11172684_

Round 1

Reviewer 1 Report

The present study was conducted to explore the binding properties of the soluble plasma proteins HRG and STC2 both to each other and to the U937 cells using the uartz crystal 32 microbalance (QCM) methodology. The biological activity of HRG and STC2 were tested by measuring its effect on the U937 cells differentiate into macrophage/monocyte-like cells after stimulation with VitD3. The authors detected that HRG  increased the proportion of phagocytotic cells in both undifferentiated and differentiated cultures. However, the phagocytosis was significantly higher in the differentiated than in the undifferentiated U937 monocytes.  This was different for the STC2 as it has no effect on the phagocytosis and attenuates the HRG’s ability to upregulate phagocytosis by U937 monocytes.

The binding of both proteins was assisted. HRG has an estimated binding affinity of 55 nM on the immobilized STC2, while the STC2 was not retained when the HRG is immobilized suggesting that the binding between the HRG and STC2 depends on the HRG s conformation or a binding pocket in the HRG that is compromised when the HRG is immobilized.

The binding affinity of both proteins to the live U937 cells, vitD3-differentiated or not were tested; individually and in complex. Binding of both HRG and STC2 to undifferentiated U937 cells was low and the binding increased when cells were induced to differentiate by vitD3 treatment. When a mixture of HRG and STC2 was tested on the differentiated U937 cells, the binding affinity remained in the µM range, similar to that recorded for the individual proteins.   

Minor Comments

1. Please change the title of the article to be more specified reflecting major finding or results of the study.

2. The authors cited previous published papers in the results section which may confuse the readers  e.g in line 249, and in line 214

Author Response

Reviewer 1, minor Comments

  1. Please change the title of the article to be more specified reflecting major finding or results of the study.

Response: The title has been changed to “Quartz-crystal microbalance measurement of Histidine-rich glycoprotein and Stanniocalcin-2 binding to each other and to inflammatory cells”

  1. The authors cited previous published papers in the results section which may confuse the readers  e.g in line 249, and in line 214

Response: The lines around 214 have been taken out and instead the lines around 249 (current lines 224-226) have been adjusted to read: “HRG is classified an intrinsically unstructured protein [9], i.e. a protein that lacks a fixed three-dimensional structure, and it may therefore be structurally less stable and in particular, not retain a more complex binding epitope upon immobilization.”

In addition, I have added the following to the introduction, lines 67-70: “The structure of HRG’s NH2-terminal domain has been solved, indicating a redox-regulated release of the NH2-terminal domain from the His/Pro-rich domain [8]. HRG has also been classified as an intrinsically unstructured protein, unable to attain an ordered or fixed conformation [9].”

Reviewer 2 Report

In this manuscript, the authors extended their previous study on the interaction between HRG and STC2. In this study, the interaction between these proteins was further characterized, and how their interaction could impact monocyte phagocytic function and binding were also investigated. Overall, the manuscript is interesting and relevant to the field, providing clear data to support their conclusions. The authors should also consider the following minor points:

1.     Can the authors please clarify how HRG could stimulate the phagocytosis of beads by monocytes?

2.     Out of interest, does proteolytic cleavage of HRG (e.g. by plasmin) or Zn2+ affect the interaction with STC2. This is beyond the scope of the current study.

3.  Regarding Line 248, can the authors please clarify the sentence “This is in agreement with…”.

4.     Regarding Figure 4, can cell fixation lead to the exposure of intracellular ligands for HRG?

5.     Again, beyond the scope of the current study, but would you expect for example proteinase K treatment could affect HRG binding to fixed U937?

Author Response

REV 2

Comments and Suggestions for Authors

1. Can the authors please clarify how HRG could stimulate the phagocytosis of beads by monocytes?

Response: The increased phagocytosis is a consequence of HRG-mediated gene regulation. Exactly how has not been investigated and is outside the scope of this study. An explanation has been added to lines 192-194 “The effects of HRG and STC2 on phagocytosis by the U937 cells are presumably a direct consequence of the gene regulatory effects of these proteins in U937 cells [16]  as well as in tumor-associated macrophages [1].”

  1. Out of interest, does proteolytic cleavage of HRG (e.g. by plasmin) or Zn2+ affect the interaction with STC2. This is beyond the scope of the current study.

Response: These are very good and important questions. The following lines have been added to the end of Discussion, lines 398-400 “Further studies include to determine whether the interaction between STC2 and HRG is Zn2+-dependent, to identify, in each protein, the minimal binding stretch for their interaction and ultimately, to identify the cell surface expressed binding proteins/receptors for HRG and STC2.”

  1. Regarding Line 248, can the authors please clarify the sentence “This is in agreement with…”.

Response: This sentence (currently lines 224-226)  has been adjusted to read ““HRG is classified an intrinsically unstructured protein [9], i.e. a protein that lacks a fixed three-dimensional structure, and it may therefore be structurally less stable and in particular, not retain a more complex binding epitope upon immobilization.”

  1. Regarding Figure 4, can cell fixation lead to the exposure of intracellular ligands for HRG?

Response: In our experience cells are not lysed upon fixation and we do not believe intracellular ligands become exposed. However, this cannot be proven and we have added to lines 306-308 “However, we cannot categorically exclude that the binding sites exposed upon fixation at least in part could represent intracellular ligands for HRG [3], either proteinase K-resistant or -sensitive.”

  1. Again, beyond the scope of the current study, but would you expect for example proteinase K treatment could affect HRG binding to fixed U937?

Response: We included Proteinase K in the comment above. We agree this should be done.